# Phenotypes of caregiver distress in military and veteran caregivers: Suicidal ideation associations

**Roxana E. Delgado**[1,2ᴑ‡*], **Kimberly Peacock**[1,2‡], **Chen-Pin Wang**[3‡], **Mary Jo Pugh**[4,5ᴑ‡]

**1** Department of Medicine, General and Hospital Medicine Division, The University of Texas Health Science Center at San Antonio (UT Health San Antonio), San Antonio, TX, United States of America, **2** Center for Research to Advance Community Health (ReACH), UT Health San Antonio, San Antonio, TX, United States of America, **3** Department of Population Health Sciences, The University of Texas Health Science Center at San Antonio (UT Health San Antonio), San Antonio, TX, United States of America, **4** Division of Epidemiology, Department of Medicine, University of Utah School of Medicine, Salt Lake City, UT, United States of America, **5** IDEAS Center of Innovation, VA Salt Lake City Health Care System, Salt Lake City, UT, United States of America

ᴑ These authors contributed equally to this work.
‡ RED and MJP are joint senior authors on this work. KP and CPW also contributed equally to this work.
* delgadore@uthscsa.edu

**Data Availability Statement:** All relevant data are within the manuscript and its Supporting Information files.

**Funding:** RED This project was supported with a pilot grant awarded by the Center for Research to

## Abstract

The United States (US) has been at war for almost two decades, resulting in a high prevalence of injuries and illnesses in service members and veterans. Family members and friends are frequently becoming the caregivers of service members and veterans who require long-term assistance for their medical conditions. There is a significant body of research regarding the physical, emotional, and social toll of caregiving and the associated adverse health-related outcomes. Despite strong evidence of the emotional toll and associated mental health conditions in family caregivers, the literature regarding suicidal ideation among family caregivers is scarce and even less is known about suicidal ideation in military caregivers. This study sought to identify clusters of characteristics and health factors (phenotypes) associated with suicidal ideation in a sample of military caregivers using a cross-sectional, web-based survey. Measures included the context of caregiving, physical, emotional, social health, and health history of caregivers. Military caregivers in this sample (n = 458) were mostly young adults (M = 39.8, SD = 9.9), caring for complex medical conditions for five or more years. They reported high symptomology on measures of pain, depression, and stress. Many (39%) experienced interruptions in their education and 23.6% reported suicidal ideation since becoming a caregiver. General latent variable analyses revealed three distinct classes or phenotypes (low, medium, high) associated with suicidality. Individuals in the high suicidality phenotype were significantly more likely to have interrupted their education due to caregiving and live closer (within 25 miles) to a VA medical center. This study indicates that interruption of life events, loss of self, and caring for a veteran with mental health conditions/suicidality are significant predictors of suicidality in military caregivers. Future research should examine caregiver life experiences in more detail to determine the feasibility of developing effective interventions to mitigate suicide-related risk for military caregivers.

Advance Community Health (ReACH) at The
University of Texas Health Science Center at San
Antonio (UT Health San Antonio), San Antonio, TX.
The funders had no role in study design, data
collection and analysis, decision to publish, or
preparation of the manuscript.

**Competing interests:** The authors have declared
that no competing interests exist.

## Introduction

Military caregivers (family and friends assisting a service member or veteran with their activities of daily living (ADL) or the instrumental activities of daily living (IADL)) have always been part of the military community, but this role has received greater visibility for veterans of post-9/11 conflicts (i.e., Afghanistan/Iraq wars) due to the improved survivability of previously fatal injuries resulting from better protective gear and the long duration of the conflicts. Military caregivers are the first line of response to the long-term home care of veterans with war-related injuries. Due to the young age of post-9/11 military caregivers, the duration of caregiving could be decades with the possibility of 50 years or more [1]. Thus, military caregivers of post-9/11 veterans may face unique challenges related to long-term health, financial, and social aspects of caregiving [2].

Theoretical models of caregiving such as the "Military and Veteran Caregiver Experience Map" [3] suggest that baseline characteristics influence the caregiver's ability to meet the new demands of caregiving, which may lead to caregiver stress/strain depending on the extent to which current identity and roles are altered by caregiving requirements. The Military and Veteran Caregiver Experience Map depicts the factors that contribute to the caregiver trajectory, the dynamics across a span of time, and events in the caregiving journey. Over time, the caregiver may shift priorities and seek help within current social/family circles or the healthcare system, and may develop new coping skills. When the caregiver is not able to shift priorities and/or obtain needed support within social relationships or the healthcare system, there is a negative impact on caregiver well-being, continued dysfunction, and diminished veteran, caregiver, and family function, which can lead to a negative impact on baseline characteristics and a negative spiral of health and well-being. When the caregiver is able to adapt/cope with new roles and responsibilities, there is a positive impact on veteran, caregiver, and family well-being and baseline characteristics with a positive trend for health and well-being.

Models of suicidality likewise suggest that suicide risk is a combination of stable and dynamic properties [4]. Suicide risk has stable characteristics that resist change over time (e.g., biological/ genetic characteristics) and dynamic characteristics that fluctuate in response to environmental and individual processes. According to the fluid vulnerability theory, suicidal behaviors emerge as a result of the interaction between dynamic and stable risk processes [4]. The fluid vulnerability theory suggests that there are factors that play an important role in the chronicity of suicide symptoms that are aggravated over time due to associated factors [5]. Given the stresses associated with caregiving (e.g., social isolation, lack of sleep, interrupting education, ending employment, changes in roles/identity), which may add to or accentuate the stable risk processes, suicidal ideation, attempt, and completion may be significantly elevated in caregivers. Indeed, more than three decades of research has shown that caring takes a significant toll on the physical health, mental health, social engagement, career prospects, sense of self, and financial security of family caregivers [6–10]. Estimates show that 40–70% of caregivers have clinically significant symptoms of depression, with approximately one quarter to one half of these caregivers meeting the diagnostic criteria for major depression [11]. Emerging literature has also found that among caregivers, significant risk factors for suicidal thoughts included being unemployed, living without a partner, having lower levels of social support, having a chronic physical disorder, a mood disorder or an anxiety disorder, and having impaired social, physical, and emotional functioning [12].

While prior studies have started addressing risk for self-harm such as suicidal ideation/ attempt, to our knowledge none of the studies have addressed this serious issue in military caregivers—a population whose young age and lengthy duration as a caregiver my put the caregiver at significant risk of suicidal ideation/attempt due to changes in mental health,

physical health, and roles/identity that occur after becoming a caregiver. To address this gap we identified phenotypes (clusters of symptoms/characteristics) of risk for suicidal ideation/attempt in a national sample of caregivers of wounded, ill, and injured veterans. We hypothesized that there would be two to four phenotypes (e.g., low-, medium-, high-risk) associated with measures of perceived stress, depression and, consistent with the fluid vulnerability theory, a higher loss of self in the caregiver since becoming a caregiver. We further hypothesized these phenotypes would be associated with suicidal ideation/attempt in military caregivers during the period since becoming a caregiver.

## Methods

Upon approval by the Institutional Review Board (IRB) at The University of Texas Health Science Center at San Antonio (UT Health San Antonio), we convened a "Military and Veteran Caregiver Advisory Group" and engaged the support of national organizations serving military caregivers for distribution of our web-based survey. The Military and Veteran Caregiver Advisory Group consisted of ten caregivers caring for a wounded, ill, and injured veteran: spouses (4) between the ages of 30 and 60 years old; parents (3) older than 55 years old caring for a veteran of the wars in Iraq and Afghanistan (post-9/11); and spouses of veterans (3) of the wars before Iraq and Afghanistan (pre-9/11). Members represented the various characteristics of military caregivers (i.e., pre-and post-9/11, length of caregiving time, myriad of injuries) across the United States [2]. The advisory group provided recommendations for the survey that helped address the diversity of caregivers in the community and assisted with the testing of the web-based survey instrument. Goals of the instrument testing were to measure the average time of completion, content, language, and order of the questions/instruments. The advisory group also helped facilitate recruiting participants from interest groups related to the advisory group's caregiving focus. Caregivers responded to web-based links in social media or e-mail information from caregiver organizations. Caregivers were not compensated for their time. To increase the response rate, survey length was limited to allow for completion within 30 minutes.

### Participants

A convenience sampling technique was used to collect data via the web-based survey from individuals who were: 1) 18 years or older; 2) self-identified as the caregiver of a wounded, ill or injured service member or veteran; and 3) proficient English speakers.

### Procedures

Surveys were disseminated via an electronic link through social media, newsletters, flyers, and e-mails from national nonprofit organizations serving this population (e.g., Elizabeth Dole Foundation, Military Veteran Caregiver Network, Hearts of Valor). Participants responded to web-based links in social media or e-mail information from caregiver organizations. Participants were not compensated for their time. To increase the response rate, survey length was limited to allow for completion within 30 minutes.

### Web-based survey

In addition to sociodemographic characteristics and the caregiving context, the web-based survey consisted of validated instruments and items measuring four main domains: 1) physical health, 2) emotional/behavioral health, 3) well-being/social development, and 4) health history.

The only identifier collected was an e-mail address to provide a mitigation strategy for those who reported prior suicidal ideation (see below).

**Sociodemographic characteristics.** The survey collected information regarding general demographics (age, sex, race, and education), the number of years as a caregiver, and total number of children.

**Context of caregiving.** Participants were first asked to describe characteristics unique to military and veteran populations (e.g., number of deployments) followed by information about their caregiving situation, including characteristics of caregiving (time, tasks, and the veteran's medical conditions with which the caregiver assists) and compensation from the Veterans Affairs (VA) Caregiver Support Program. Participants were also asked to identify the types of conditions for which they provided care (e.g., amputation, burn, TBI, PTSD, depression).

**Physical health.** Physical health status was measured using the Quality of Life, General Health Questionnaire, also known as the VR-12 [13], which measures health-related quality of life in the domains of general health perceptions, physical functioning, role limitations due to physical and emotional problems, bodily pain, energy-fatigue, social functioning, and mental health. The VR-12 is calculated using an algorithm developed by Selim et al. [13] with normative values of 50 (SD 10) on each scale with higher numbers indicating a more positive self-reported health status. Research has found that a one-point increase on the VR-12 is associated with lower health expenditures [14].

Pain was measured with the six-item Pain Impact Questionnaire (PIQ-6™ or "PIQ-6") [15], a patient-based assessment designed to measure pain severity and the impact of pain on an individual's health-related quality of life (HRQOL). Each item is rated on a six-point Likert scale ranging from "none" to "very severe." Scores on the PIQ-6 of 58 or higher indicate pain that should be assessed and treated by a medical professional, with scores 64 and higher indicating severe impact [16].

**Emotional/Behavioral health domain.** Several aspects of emotional well-being were evaluated. Depression symptoms were measured using the Patient Health Questionnaire (PHQ-9), which is used in primary care to screen for depression [17]. The score ranges from 0 to 27, with five levels of depression: 1) normal (0–4); 2) Mild (5–9); 3) Moderate (10–14); 4) Moderately severe (15–19); and 5) Severe (20–27) [17]. Stress was measured using the 10-item Perceived Stress Scale (PSS), which measures how unpredictable, uncontrollable, and overloaded respondents find their lives [18, 19]. Normative scores for the PSS are 13.7 (SD = 6.6) for women and 12.1 (SD = 5.9) for men [19].

Participants were also asked about caregiver suicidal ideation using the question, "Since becoming a caregiver, have you thought of harming yourself or trying to take your own life?" based on the Assessment of Suicidal Ideation and Plan [20, 21]. Participants who reported previous and current suicidal ideation were contacted by the research team by e-mail and provided a referral to mental health programs that serve military caregivers and a resource guide for additional services in the participant's community.

**Social function/Well-being domain.** The 16-item Caregiver Well-Being Scale short-form [22] measures basic needs (BN) and activities of daily living (ADL). The BN items represent biological, psychological, and social needs and the ADL items represent ways to meet these needs. Scores for BN and ADL each ranged from 0–5, with a higher score indicating better social function and well-being [22].

In addition, caregivers may experience a change in the way they perceive themselves socially, prompting the measurement of the participant's sense of self-loss. The Loss of Self instrument is a two-item questionnaire that measures the extent to which the caregiver reported a self-loss due to caregiving and engulfment resulting from being consumed by the

caregiving role [23]. The two questions were: How much have you lost: a) a sense of who you are and b) an important part of yourself? Each item is measured with a four-point Likert scale ranging from "not at all" (1) to "completely" (4).

**Caregiver health history.** We identified previously diagnosed health conditions by asking caregivers to provide information about their health history and identify conditions for which they received a diagnosis by a healthcare provider during the period of time since beginning their caregiver role. We included the following medical conditions as part of the health history: anxiety, insomnia, autoimmune disorders, and migraines/headaches.

## Data analysis

We conducted a descriptive analysis of the sociodemographic characteristics of the caregiver, context of caregiving, and health outcomes from each of the four domains, including the point prevalence of medical conditions reported by the caregivers at the time of the survey.

After examining the distribution of measures across the four domains (cognition, emotion, physiology, and behavior), we conducted general latent variable modeling (GLVM) to identify distinct caregiver distress phenotypes based on scores from self-report measures of caregivers (PSS, ADL, burn, PIQ, aLOS of self, bLOS of self, years of care, VR12) and conditions for which they provided care (ALS, depression, PTSD, suicide ideation, TBI, and trauma), adjusting for the covariates associated with the clusters.

GLVM is a robust parametric modeling technique used to identify distinct unobserved subgroups within a population based on a mixed type of multivariate outcomes (continuous and categorical) such that individuals of the same latent class share a similar joint distribution of these outcomes. Each class identified by GLVM is characterized by a distinct pattern of means and variances associated with the continuous outcomes and frequencies associated with the categorical outcomes. The GLVM consists of: (i) pre-specifying the number of latent classes; (ii) conditioned on each latent class, modeling the joint distribution of continuous and categorical outcomes with varying model parameter estimates to differentiate between classes; and (iii) predictors associated with class membership were identified based on the modified Bolck, Croon, and Hagenaars method [24], which accounts for classification errors and corrects underestimation of predictor effects. In the GLVM, covariates of class membership included distance from residence to VA care, interruption of education, and days per week of caregiving for caregivers. Each GLVM was run using Mplus 8.2 allowing 20 different start values to ensure global maximization of the model estimates and that the models produced stable results regardless of the start value. The best fitting GLVM was identified primarily using Bayesian information criterion (BIC): models with smaller BIC values indicate a better fit. The GLVM for this study explored two to four classes since the model fit did not improve nor converge when assuming four or more classes. In addition to goodness of fit, the clinical relevance of the models was evaluated, interpreting the meaning of classes as *caregiver distress phenotypes*.

Logistic regression analysis then examined the odds of suicide ideation by caregiver distress phenotype controlling for age, time (hours) providing daily care, interruption in the education, and distance to the VA.

## Results

### Sample characteristics

Table 1 provides the descriptive characteristics of the sample. Of the participants that were screened (N = 502), 93% (n = 458) were eligible and completed the survey between April and August 2017. Approximately half (56.3%) were under the age of 40 caring for veterans of similar age (*M* = 40.8, *SD* = 9.9); most had at least one combat deployment (93.3%). The majority

**Table 1. Characteristics of military and veteran caregivers.**

| Characteristics of Military and Veteran Caregivers (N = 458) | N | % | Missing n (%) |
|---|---|---|---|
| **Caregiver Relationship to Veteran** | | | 23(5.0) |
| Spouse | 400 | 87.3 | |
| Parents | 9 | 2.0 | |
| Other (sibling, grandparent, friend) | 26 | 5.7 | |
| **Education** | | | 29(6.3) |
| High school diploma or equivalent | 39 | 8.5 | |
| Trade vocational school | 26 | 5.7 | |
| Some college but no degree | 138 | 30.1 | |
| Associates degree | 69 | 15.1 | |
| Bachelors | 96 | 21.0 | |
| Postgraduate | 48 | 10.5 | |
| Doctoral degree | 6 | 1.3 | |
| Other | 3 | .7 | |
| | **Mean** | **SD** | |
| Age | 39.8 | 9.9 | 9(2.0) |
| **Caregiver Health-Related Outcome Measures** | | | |
| Pain* (*Pain Inventory Questionnaire-6*) | 56.4 | 9.2 | 42(9.2) |
| Depression** (*Patient health Questionnaire-9*) | 11.3 | 6.4 | 69(15.1) |
| Stress** (*Perceived Stress Scale*) | 21.6 | 4.2 | 84(18.3) |
| Well-being** (*The Caregiver Well-being Scale*) | 3.0 | .6 | 87(19.0) |
| Quality of Life (*Veterans RAND 12*) | 38.1 | 3.0 | 47(10.3) |
| Loss of Self** (*Loss of Self*) | 2.7 | .8 | 74(16.2) |
| Alcohol use (*Alcohol Use Disorders Identification Test_C*) | 2.0 | 1.7 | 255(55.7) |
| **Caregiver Self-Reported Health and Diagnosis** | **n** | **%** | **Missing n (%)** |
| Anxiety | 239 | 52.2 | |
| Insomnia | 124 | 27.1 | |
| Migraine/headaches | 136 | 29.7 | |
| **Context of Caregiving** | | | |
| Received Financial Compensation for Caregiving (VA Caregiver Stipend) | 220 | 48.0 | 5(1.1) |
| **Veteran (Care Recipient) Health History** | | | N/A |
| ALS | 17 | 3.7 | |
| Amputation | 16 | 3.5 | |
| Blindness | 7 | 1.5 | |
| Burn | 8 | 1.7 | |
| Chronic pain | 335 | 73.1 | |
| Depression | 367 | 80.1 | |
| Epilepsy/Seizures | 48 | 10.5 | |
| Posttraumatic stress disorder (PTSD) | 409 | 89.3 | |
| Lung Conditions | 58 | 12.7 | |
| Skin | 113 | 24.7 | |
| Spinal Cord Injury | 77 | 16.8 | |
| Suicidal ideation | 137 | 29.9 | |
| Traumatic brain injury (TBI) | 313 | 68.3 | |

Note: Differences between the phenotypes

**p< = 0.001

*p < .01.

Nearly a quarter (23.6%) reported suicide ideation since becoming a caregiver.

of the participants were Caucasians (84.4%); there were also Hispanics (8.5%), Asian/Pacific Islanders (1.8%), African Americans (1.8%), and identified as other (3.5%). Nearly a third of the participants had completed four or more years of college and 39% reported interruption in their education due to their caregiving role, mostly while pursuing a bachelor's degree. On average, participants were married for 12.5 years (SD = 8.5) and the majority (60.4%) had children 18 years old and younger.

Most participants were the care recipient's spouse (Table 1). Over 60% of caregivers had been providing care for five or more years, with 14% providing care for over 11 years. The majority (60.9%) reported caring for veterans with five or more medical conditions. Caregivers experienced poor health-related quality of life as measured by the VR-12 [13] (Table 1). Pain was highly prevalent in the sample, with more than half meeting the criteria for pain requiring clinical intervention [16]. A large number of participants (84.1%) exhibited depression symptoms as measured by the PHQ-9, with 29.6% exhibiting moderate to severe depression [17]. Perceived stress [18] among caregivers was also high and scores on measures of well-being and loss of self were low compared to population/prior sample norms [22, 23]. Over half had been diagnosed with depression and nearly 30% were previously diagnosed with insomnia or migraines/headaches. Table 1 also shows that the health conditions most common in care recipients were depression, chronic pain, post-traumatic stress disorder (PTSD), traumatic brain injury (TBI), and suicidal ideation.

### General latent variable model analysis

GLVM analysis found that among the GLVMs that converged, a three-class solution had the best fit based on the lowest BIC and clinically meaningful interpretation (Table 2, Figs 1 and 2).

**Table 2. Caregiver distress cluster/class response.**

| | | Class for Caregiver Distress | | |
|---|---|---|---|---|
| | | Low n = 113 (24.7%) | Medium n = 251 (54.8%) | High n = 94 (20.5%) |
| **Caregiver health-related characteristics** | Perceived Stress | 19.82 | 21.38 | 23.86 |
| | Quality of Life- Activities of Daily Living | 3.52 | 3.04 | 2.61 |
| | QoL Basic Needs (biopsychosocial needs to sustain life) | 3.74 | 2.80 | 2.12 |
| | Pain (PIQ-6) | 51.75 | 56.32 | 61.05 |
| | Loss of Self- "A sense of who you are" | 2.18 | 2.62 | 3.53 |
| | Loss of Self- "An important part of yourself" | 2.14 | 2.62 | 3.55 |
| | Well-being "feel good" | 3.79 | 2.66 | 1.17 |
| | Well-being "finance" | 3.57 | 2.28 | 1.60 |
| | Depression Symptoms (PHQ-9) | 5.12 | 11.21 | 17.79 |
| | Anxiety | 0.25 | 0.55 | 0.76 |
| | Insomnia | 0.10 | 0.28 | 0.44 |
| | Migraines/Headaches | 0.15 | 0.32 | 0.40 |
| **Proportion of caregivers caring for a Veteran with...** | ALS | 0.06 | 0.03 | 0.04 |
| | Depression | 0.53 | 0.88 | 0.92 |
| | Posttraumatic stress disorder | 0.69 | 0.97 | 0.95 |
| | Spinal cord injury | 0.14 | 0.19 | 0.16 |
| | Veteran suicide risk behavior | 0.10 | 0.28 | 0.57 |
| | Traumatic brain injury | 0.48 | 0.74 | 0.77 |
| | Amputation/Burn | 0.05 | 0.10 | 0 |

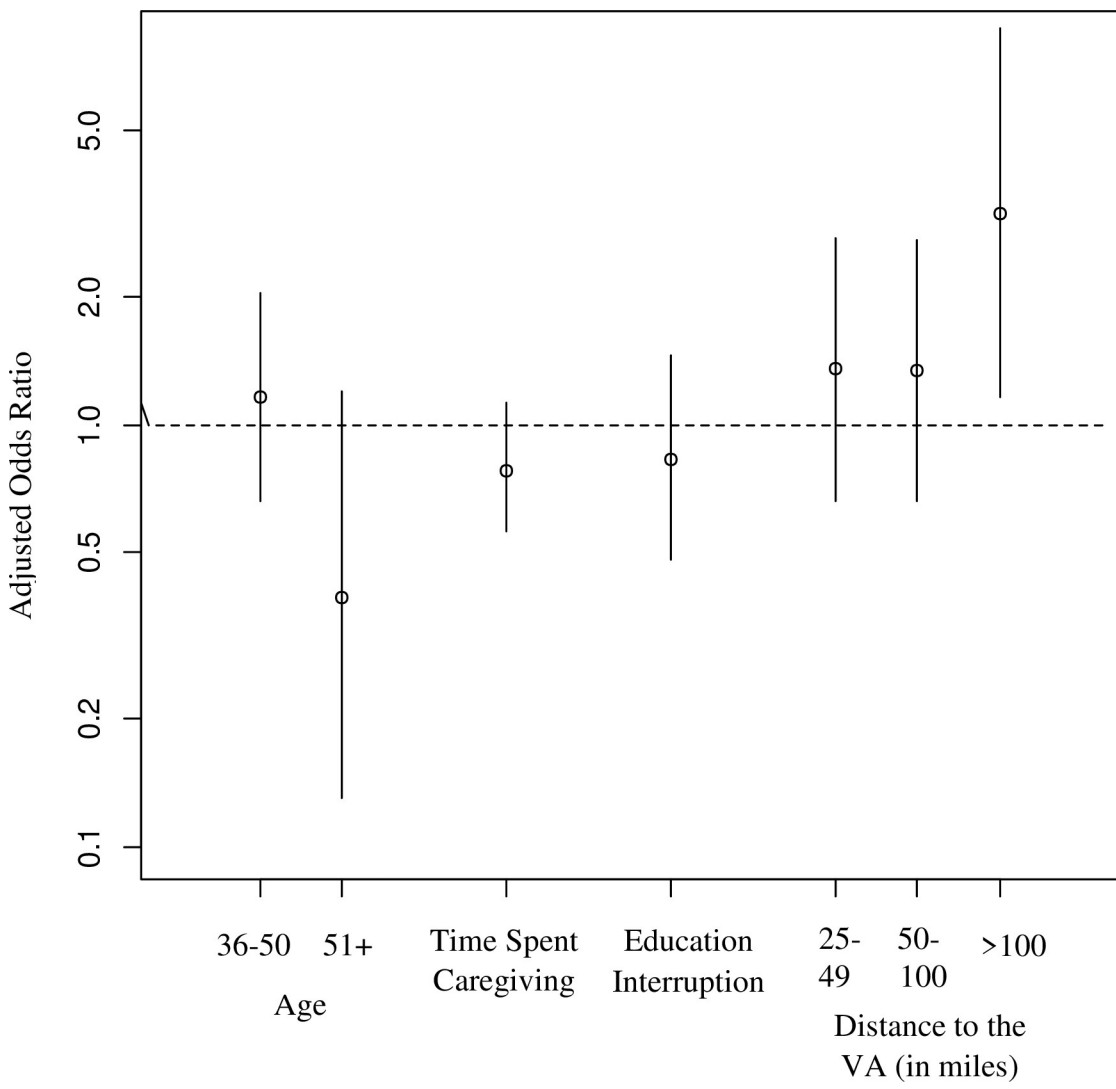

**Fig 1. Adjusted odds ratio of caregiver distress covariates.**

The three phenotypes identified by GLVM reflected high, medium, and lower distress. The high-distress phenotype (20% of the cohort) had the highest reported levels of stress, depression, pain, loss of self, and the highest prevalence of previously diagnosed anxiety, depression, insomnia, and migraines. The high-distress phenotype also reported the lowest levels of well-being and financial security.

The medium distress phenotype (55% of the cohort) had patterns similar to the high-distress phenotypes, but with levels that were significantly less extreme than those of the high-distress phenotype. Finally, the low-distress phenotype (25% of the cohort) had the lowest (or highest for well-being/financial security) scores of all groups, except for reporting the highest probability of potentially high-risk alcohol use compared to those in the medium- and high-distress phenotypes. Caregiver distress phenotypes were also associated with conditions for which care was provided, but the relationships of differences were not as seemingly linear as was found for caregiver measures (Table 2). For example, caregivers in the low suicide risk behavior (SRB) phenotype had far lower probabilities of caring for depression and PTSD than

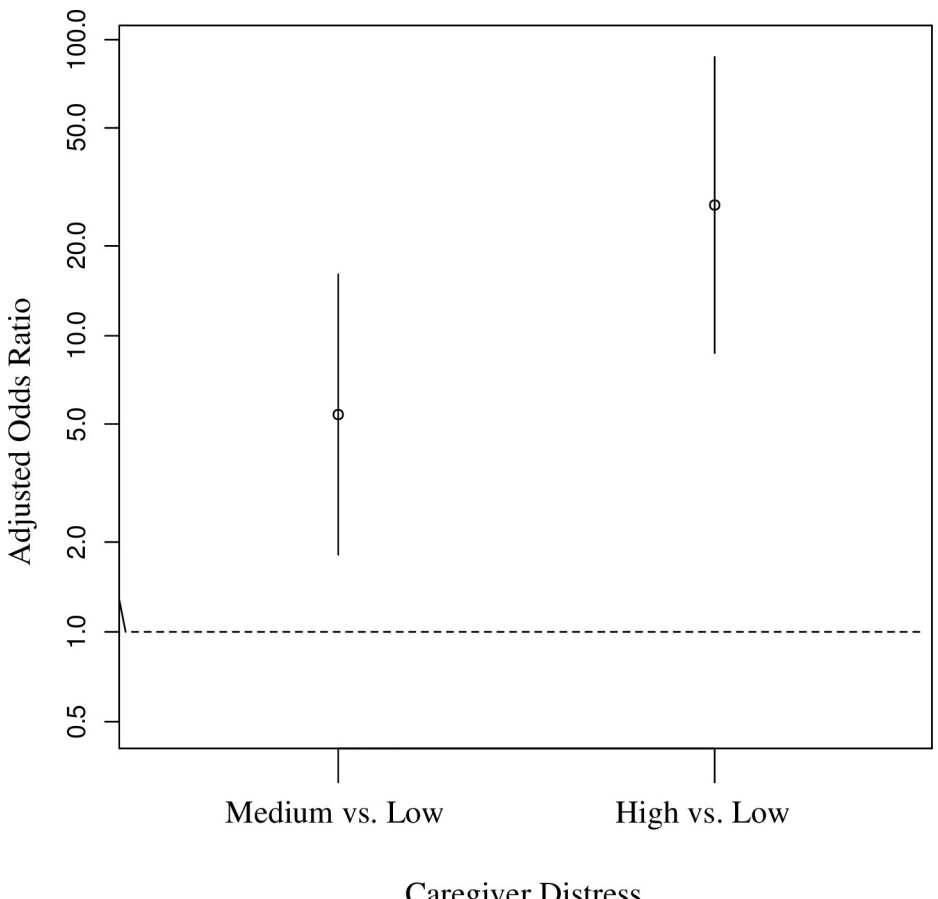

**Fig 2. Adjusted odds ratio of caregiver distress classes.**

the medium and high SRB risk phenotypes, respectively. Caring for a veteran with suicidal ideation was one variable, however, that had a more linear relationship on probabilities in medium and high SRB risk phenotypes. Conversely, the probability of caring for more physically focused (visible) conditions (e.g., ALS, amputation/burn injuries) was significantly higher for the low-distress phenotype compared to the medium- and high-distress phenotypes. Covariates significantly associated with high-distress phenotype were interruption of education and shorter travel distance to a VA medical facility. The distress phenotypes of caregivers that reported being part of the VA caregiver program were not different, $X^2$ (2, N = 456) = 5.90, $p$ = .06 than those who were not part of the program.

Based on the Bolck, Croon, and Hagenaars method [24], the estimated proportions of caregiver suicidal ideation were 6%, 20%, 50% for the low-, medium-, and high-risk phenotypes (p-value <0.001 based on chi-square test), respectively. Logistic regression analysis adjusted for covariates found that individuals in the medium- and high-distress classes were significantly more likely than those in the low-distress class to report prior suicidal ideation (adjusted odds ratios [AOR] 5.40 [95% confidence interval 1.82–16.04] and 27.64 [95% confidence interval 8.76–87.25; all p < .01], respectively). The AUC associated with this caregiver suicidal ideation prediction model was 0.779 (95% CI 0.73–0.84).

## Discussion

Prior studies have described the association of caregiving to individual symptoms such as depression, anxiety, and hopelessness; and these characteristics have all been linked to suicide ideation and suicide-related behaviors in the general caregiver population [12, 25, 26]. This study is the first, to our knowledge, to develop computational phenotypes of caregiver distress using diverse measures of physical, social, and emotional well-being. We found high-, medium- and low-distress phenotypes that were significantly associated with suicide ideation since becoming a caregiver. The GLVM examining the caregiving context, mental health, physical health, and social function/well-being revealed three distinct phenotypes that were strongly associated with suicidal ideation. These caregiver distress phenotypes were character-ized by seemingly linear associations in risk predictors such that the high-distress phenotype exhibited significantly worse emotional outcomes, followed by medium-risk and low-risk. Studies have shown the emotional toll of caregiving, especially those caring for mental health conditions, significantly increases the caregiver's risk for suicidal behavior [26–28]. The high prevalence of mental health conditions, polytrauma, and comorbidities among veterans may worsen the mental health-related outcomes in their caregivers. In fact, caregivers assisting individuals with mental health conditions or psychiatric disorders are more prone to experiencing a negative impact on their emotional well-being [28, 29]. This was evident in the medium- and high-caregiver distress phenotypes showing a high probability of suicide idea-tion among caregivers of veterans with behavioral health conditions. The long-term care of veterans with complex medical conditions and mental health comorbidities may pose addi-tional risks for suicidal thoughts among their caregivers. Suicidal ideation rates in this sample of military caregivers were higher than rates in other published studies measuring suicide-related behavior among family caregivers [26, 30, 31].

There is limited existing literature on suicide in caregivers in the general population, and it primarily focuses on caring for a loved-one with mental health conditions, Alzheimer, or Dementia. When compared with studies addressing caregiving and suicide in the general pop-ulation, this sample of military caregivers exhibited higher rates of suicidal ideation. This marked difference could be attributed to various factors, including that military caregivers per-form a higher complexity of care at a younger age, during the prime of their adulthood [32–34]. We recommend additional epidemiological studies to identify and fully understand care-giving and suicide in military and civilian caregivers. When compared to other studies that examined suicide-related behavior among family caregivers, anxiety and depression were also contributors to adverse exacerbation of the caregiver's emotional well-being. Similar findings were reported in studies of caregivers assisting family members with dementia, cancer, and disability [25, 31]. Our sample exhibited a higher risk of suicidal ideation among those with depression and perceived stress. Additionally, unemployment and lack of social support are factors associated with suicide ideation in caregivers [12].

Often, when a person acquires a disability during adulthood some years of productive work are lost. Similarly, when a family member assumes the role of caregiver, a sudden interruption of the caregiver's education and/or career goals can occur, with a resulting loss of productive years. Post-9/11 caregivers, such as many of those in this study's sample, were young to mid-dle-aged adults, many of whom stopped their education and/or their professional endeavors to care for their veteran. This study found that individuals in the high- and moderate-distress phenotypes had very high odds of suicide ideation since beginning the caregiving role. Studies of the long-term impact on productivity, health, and adverse outcomes such as suicide ideation are needed. However, future research should quantify the possible impact of caregiving using an approach similar to the disability-adjusted life years (DALY), which represents the gap

between the actual health status and an ideal health situation where the entire population lives to its life expectancy with no disease and disability [35–37].

Loss of self has been studied in the context of the patient with some evidence of the impact in cancer survivors [38] and chronically ill patients [39] and highly associated to self-esteem, but loss of self has rarely been examined in caregiver research. Extant studies have found loss of self mostly among caregivers that are spouses, younger, and women [23]. Factors like interruption in career and school resulting from an engulfment in the caregiver role, may be contributors to this sense of identity and loss. This study found that high- and moderate- distress phenotypes had scores indicating the highest loss of self, the highest probability of interrupting education to provide care, and the highest odds of suicide ideation. It is possible that the complex interaction of interrupting education, loss of self, and possible changes in the relationship are associated with a mismatch between the actual and ideal self, which is then associated with ambiguous loss or alteration in identity [32–34, 40].

While research has examined the concept of ambiguous loss [40–42] among care recipients that have suffered an injury or illness, the self-loss of the caregiver and its association to their health and well-being merits further exploration. Ambiguous loss explains the grieving process of family caregivers of people with dementia, Alzheimer's disease, TBI, and other neurological disorders [43, 44], where the changes and perceived loss are for the care recipient, but not as a caregiver loss that pertains to the caregiver's essence and identity. Additional research is needed to understand the complex relationships and mediating factors that contribute to changes in identity and health outcomes in a population of military caregivers. Research is also needed to identify the most appropriate interventions and timing of those interventions to optimize health and well-being of this population of caregivers.

The travel time to the nearest VA facility and its association to the high stress phenotype may relate to confounding by indication in that people caring for veterans with more severe mental health symptoms (e.g., depression/suicidality) may want to live closer to a VA to make it easier to obtain urgent care from clinicians who are familiar with their care rather than a community emergency room where continuity of care may not be available. The increased burden of this care was demonstrated in the distress phenotypes, which is one possible explanation for this finding (this finding is further discussed in the methods section). Why caregivers in the low-distress group were more likely to report risky alcohol use is less clear. We hypothesize that alcohol use may be a primary stress management approach that leads to less perceived stress, less depressive symptoms, and lower loss of self in addition to lower risk for suicide ideation or attempt.

The limitations of this study included a convenience sample with self-reported measures. The potential selection bias that resulted from the study eligibility requirement for participants to self-identify as a caregiver was mitigated by the development of an internal algorithm that identified the responses provided and compared them to unique characteristics meeting the definition of caregiving. The participants of this study self-identified as a military and veteran caregiver. Each survey was evaluated based on the information provided regarding the caregiver role, performed tasks, period of time of caregiving as well as the care recipient's military service and medical history. These variables served as verification of caregiving status, specifically as a caregiver of a wounded, ill, and injured military or veteran. In addition, comparison of the characteristics in this study's sample to those reported by the RAND study [2] (average age, education, and health-related outcomes such as mental health symptoms of depression, stress, and anxiety) suggest that they are remarkably similar. To date, the RAND study is the only population-based data in military and veteran caregivers, which has been established in the literature as significantly different from caregivers in the general population [45]. A comparison of the characteristics in this study's sample to those reported by the RAND study

(average age, education, and health-related outcomes such as mental health symptoms of depression, stress, and anxiety) suggest that they are remarkably similar. However, the RAND study did not study suicide ideation among caregivers; and with no other studies exploring the topic of suicidal ideation in military and veteran caregivers, we cannot compare this study's results with a similar sample of caregivers. Minority groups in the sample of this study were underrepresented. This is a limitation that could have resulted from various factors, including recruitment strategies. This limitation may play a role in the rate of behavioral health conditions reported in the sample, typically being higher in minority groups. The study did not collect information regarding the caregiver's pre-existing conditions, especially those associated with behavioral health that could have an association with suicidal ideation.

This study's findings identify a public health concern and a call to action. Suicide is the tenth leading cause of death in the US and suicide among veterans is already a public health problem. This study also identifies significant suicide risk for military caregivers. Presently, there are no registries or efforts in monitoring suicide among military families. Military caregivers of the latest era of war are younger, in the prime of their adult life, not yet at the age of retirement, and not yet at the age when a caregiver role is assumed, which typically occurs at the age of retirement. As a result, the emotional, physical, and social toll of caregiving in this group of younger caregivers may last for decades. A healthy and productive life while being a caregiver may require significant support from programs and policies that can strengthen the health and well-being of caregivers, with an emphasis on career progression. To date few programs are addressing suicide in military and veteran caregivers. This study sheds light on some of the factors and caregiver characteristics associated with suicidal ideation, a contribution we hope can be incorporated in future prevention initiatives. This study suggests that education interruption, loss of self, and the stress of caregiving are important contributors that warrant further evaluation.

## Supporting information

**S1 Dataset. De-identified limited dataset.**
(SAV)

**S1 File. Latent class analysis for Figs 1 & 2.**
(R)

## Acknowledgments

### Disclaimer

**Military and Veteran Caregiver Advisory Group:** Rosalinda Babin, Melissa Comeau, Emily Emmons, Dr. Precious Goodson, Alma Hall, Liz Rotenberry, Annette Ruiz, Nikki Stephens, Catherine Stobie, and Mary Ward.

**Organizations:** Elizabeth Dole Foundation, Red Cross Military and Veteran Caregiver Network, TBI Warrior® Foundation, Hearts of Valor, Yellow Ribbon Fund, and Tri-Service Nursing Research Program Family Interest Group.

Dr. Pugh also receives funding from VA Health Services Research and Development Service grant (RCS 17–297; Mary Jo Pugh PI). Any opinions, findings, conclusions, or recommendations expressed in this publication are those of the author(s) and do not necessarily reflect the views of the U.S. Government, or the U.S. Department of Veterans Affairs, and no official endorsement should be inferred.

## Author Contributions

**Conceptualization:** Roxana E. Delgado, Kimberly Peacock, Mary Jo Pugh.

**Data curation:** Roxana E. Delgado.

**Formal analysis:** Chen-Pin Wang.

**Funding acquisition:** Roxana E. Delgado, Kimberly Peacock, Chen-Pin Wang, Mary Jo Pugh.

**Investigation:** Roxana E. Delgado, Kimberly Peacock, Mary Jo Pugh.

**Methodology:** Roxana E. Delgado, Kimberly Peacock, Chen-Pin Wang, Mary Jo Pugh.

**Project administration:** Roxana E. Delgado, Kimberly Peacock.

**Resources:** Roxana E. Delgado, Mary Jo Pugh.

**Supervision:** Roxana E. Delgado, Mary Jo Pugh.

**Visualization:** Roxana E. Delgado.

**Writing – original draft:** Roxana E. Delgado, Chen-Pin Wang, Mary Jo Pugh.

**Writing – review & editing:** Roxana E. Delgado, Kimberly Peacock, Mary Jo Pugh.

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
