## [Decision Letter · Decision Letter 0]

1 Mar 2021

PONE-D-21-01599

Phenotypes of caregiver distress in military and veteran caregivers: suicidal ideation associations

PLOS ONE

Dear Dr. Delgado,

Thank you for submitting your manuscript to PLOS ONE. After careful consideration, we feel that it has merit but does not fully meet PLOS ONE’s publication criteria as it currently stands. Therefore, we invite you to submit a revised version of the manuscript that addresses all the points raised during the review process.

We look forward to receiving your revised manuscript.

Kind regards,

Gianluigi Forloni

Academic Editor

PLOS ONE

2. We note that Figure 1 in your submission contain copyrighted images. All PLOS content is published under the Creative Commons Attribution License (CC BY 4.0), which means that the manuscript, images, and Supporting Information files will be freely available online, and any third party is permitted to access, download, copy, distribute, and use these materials in any way, even commercially, with proper attribution. For more information, see our copyright guidelines: http://journals.plos.org/plosone/s/licenses-and-copyright.

(1) You may seek permission from the original copyright holder of Figure 1 to publish the content specifically under the CC BY 4.0 license.

3. Please upload a new copy of Figure 1 as the detail is not clear. Please follow the link for more information: https://blogs.plos.org/plos/2019/06/looking-good-tips-for-creating-your-plos-figures-graphics/" https://blogs.plos.org/plos/2019/06/looking-good-tips-for-creating-your-plos-figures-graphics/

Reviewers' comments:

Reviewer's Responses to Questions

**Comments to the Author**

1. Is the manuscript technically sound, and do the data support the conclusions?

Reviewer #1: Yes

Reviewer #2: Yes

2. Has the statistical analysis been performed appropriately and rigorously? 

Reviewer #1: Yes

Reviewer #2: Yes

3. Have the authors made all data underlying the findings in their manuscript fully available?

Reviewer #1: Yes

Reviewer #2: Yes

4. Is the manuscript presented in an intelligible fashion and written in standard English?

Reviewer #1: Yes

Reviewer #2: Yes

5. Review Comments to the Author

Reviewer #1: This is an interesting and well written manuscript focusing on determining phenotypes of distress among caregivers for veterans. The topic is an important one and the method used including sample size is sound. The authors also apply a sophisticated statistical approach to developing the phenotypes. Overall, the manuscript was strong and I have no major critiques. I provide a few thoughts and suggestions below, in order of appearance in the manuscript, to help strengthen and clarify the manuscript.

On line 84 you introduced the fluid vulnerability theory but did not afterwards connect it with your following sentences. Why was it introduced? Does it help clarify the reasons for increased vulnerability among veteran caregivers? Please clarify the connection or remove the reference to the theory.

Check the whole manuscript for copyediting issues and consistency. I made a few notes of errors I noted but I don’t think this list is exhaustive - On lines 92, 96, 249 and 250 the periods should be before the superscript numbers. Paragraph starting on line 81 is not indented while others are. similarly there was a typo on line 60: “veteran’s” should be ‘veterans’ Please add a comma between the ‘17’ and ‘18’ in line 166. Please remove the space before the period in line 236

I was unclear why the paragraph beginning on line 98 was part of the introduction. It describes the particulars of what you did in this study and your statistical approach - shouldn’t it be in the method section?

The sentence related to line 137 was unclear regarding if the ‘number of deployments’ was the only characteristic assessed “unique to military and veteran populations”. If so, please change “characteristics” “characteristic” and if not please add an e.g. before “number of deployments”.

Was the survey anonymous? I assumed it did not collected identifying information as collecting names/email addresses was not stated anywhere (as is common with web based surveys). However I was unsure how individuals who reported “previous and current suicide ideation” (see line 171-172) were contacted. Please clarify this in the method section. Furthermore, I assume but am not sure if you meant “previous and/or current suicide ideation” based on your current language it appears that they would need both previous and current ideation to be contacted while those with just current would not be contacted. I am not sure this was your intent. Also, it may be useful to add the number of individuals contacted for this reason.

Was the ‘loss of self instrument’ validated? Due to its brevity I would appreciate more information about this instrument including other usage and perhaps test-retest if available?

What does it mean when you “identified the extent to which caregivers had been diagnosed” on line 191. Does that mean you asked them if they had such a diagnosis? I”m not sure ‘extent’ is the best word in this context.

On line 318-319 - “Suicidal ideation rates in this sample of military caregivers were higher than rates in other published studies measuring suicide-related behavior among family caregivers”

Please add a discussion about this - do you think this was pure chance or a real effect? Perhaps a change over time or due to different research methods? might this be due to your sampling method? What was stated in the advertisements that may affect this?

Regarding the explanation beginning on line 347, there are many studies showing how identity disturbances can be related to developing suicidality. It might be worth using/citing some here.

On line 364 you state that “ the potential selection bias that resulted from the study eligibility requirement for participants to self-identify as a caregiver was mitigated by the development of an internal algorithm that identified the responses provided and compared them to unique characteristics meeting the definition of caregiving.” I did not understand what this meant. What is this internal algorithm? How was it used to help with potential selection bias? I don’t remember seeing this in your method. Please add a full explanation of this to your method section.

I also was unsure your other reason to limit the issue of selection bias was presented in a satisfactory fashion. While your study sample characteristics may have been similar to the RAND study they did not match other studies levels of suicide ideation (a central part of your results and discussion). You should note this in the context of how it may relate to selection bias as a limitation. Personally, I do not find this to be a major methodological issue or concern but this issue should be noted and left to the reader to determine if they consider this a critical limitation regarding the interpretation of the suicide findings.

Line 381 - I found the sentence beginning with ‘existing programs’ confusing. The two parts do not seem linked well. Also there was a typo - please remove the capital ‘I’

Finally, when reading the results I was hoping for a discussion of some of the more unexpected findings such as why are shorter travel distances to a VA medical facility associated with high distress phenotype and why risk for alcohol usage was associated with the lower distress phenotype. Please add some explanation for these findings to the discussion.

When reading table 1, I was unable to determine what the stars and p-values indicated. Ie what analyses were run on which variables that led to the p-values. Please clarify this

Figures need APA style labels - Figures are also unclear with text illegible. I was unable to read most of the text in the first figure and some of the text in the latter two figures.

Reviewer #2: This study is one of the first studies to examine suicide symptoms in military care giving population; highlighting a significant problem that has gained minimal attention.

The authors conducted an online survey targeted to military caregivers and reached approximately 400 individuals. Using GLVM, they determined 3 phenotypes of caregiver profile contributing to suicidal ideation.

The article is well written, and background section for the most part is synthetic and detailed.

The following items impacted my enthusiasm and if addressed will strengthen the manuscript.

1) Theoretical model: The model (figure 1) for caregivers experience map looks like a flyer and does not particularly correspond to the abbreviated description in the article. The figure is much more detailed with content that is not elaborated elsewhere and it is not clear how the choice of survey parameters is derived from this.

2) Survey instrument: The survey is anonymous and not particularly well described. How was it developed?

Was it beta tested? An internal algorithm is mentioned that “identified the responses provided and compared them to unique characteristics meeting the definition of caregiving” Please elaborate what this is and how it works. How does the survey know if the caretaking is of a Veteran or military personnel?

While the survey is anonymous, there is outreach if the participant notes suicidal ideation, past or present. How is this accomplished if survey is anonymous?

3) Participant concerns:

Given that the recruitment includes VA Caregivers Program participants (48%), these individuals are often compensated for their involvement with the Veteran. This seems fundamentally different than caregivers who do not receive financial support. How is this captured in your design and findings?

4) The participants are 85% Caucasian and <2% African American. This does not represent Veteran population nor I suspect Veteran caregivers. This is a limitation and should be mentioned. Is this due to recruitment outreach methods? What about bias of who is willing to complete survey with no compensation?

5) Statistics figures are hard to read and poor quality.

6) Limitations: The study focuses solely on symptoms coincident with caretaking responsibilities and therefore there is no data on pre-existing problems, or even prior suicidal symptoms. The introduction refers to the fluid vulnerability theory making the argument for static (e.g. biological) and dynamic processes, but this paper ignores pre-existing conditions. This needs to be acknowledged in the limitations section. What was the rationale for not asking about previous suicide symptoms/history, given the strong connection about past suicidal symptoms being significant risk factor for repeated occurrence?

7) Typo – line 382

6. PLOS authors have the option to publish the peer review history of their article (what does this mean?). If published, this will include your full peer review and any attached files.

Reviewer #1: No

Reviewer #2: No

---

## [Author Response · Author response to Decision Letter 0]

30 Apr 2021

Dear Academic Editor and Review Committee,

Thank you for your consideration and thorough review of the manuscript entitled “Phenotypes of caregiver distress in military and veteran caregivers: suicidal ideation associations.” We appreciate the opportunity to revise the manuscript. The feedback from the reviewers was very helpful and we addressed each of the recommendations as follows:

Reviewer #1

This is an interesting and well written manuscript focusing on determining phenotypes of distress among caregivers for veterans. The topic is an important one and the method used including sample size is sound. The authors also apply a sophisticated statistical approach to developing the phenotypes. Overall, the manuscript was strong and I have no major critiques. I provide a few thoughts and suggestions below, in order of appearance in the manuscript, to help strengthen and clarify the manuscript.

1. On line 84 you introduced the fluid vulnerability theory but did not afterwards connect it with your following sentences. Why was it introduced? Does it help clarify the reasons for increased vulnerability among veteran caregivers? Please clarify the connection or remove the reference to the theory. 

Response: We have clarified that a number of situations occur after military caregivers assume the role of caregiving are consistent with fluid vulnerability such as interrupting education, ending employment, and changes in role/identity, which led to increased vulnerability for caregivers. We included the following sentence:

“The fluid vulnerability theory suggests that there are factors that play an important role in the chronicity of suicide symptoms, aggravated over time due to associated factors”

Added reference: Bryan, C. J., Butner, J. E., May, A. M., Rugo, K. F., Harris, J. A., Oakey, D. N., Rozek, D. C., & Bryan, A. O. (2020). Nonlinear change processes and the emergence of suicidal behavior: A conceptual model based on the fluid vulnerability theory of suicide. New Ideas in Psychology, 57. https://doi-org.libproxy.uthscsa.edu/10.1016/j.newideapsych.2019.100758

2. Check the whole manuscript for copyediting issues and consistency. I made a few notes of errors I noted but I don’t think this list is exhaustive - On lines 92, 96, 249 and 250 the periods should be before the superscript numbers. Paragraph starting on line 81 is not indented while others are. similarly there was a typo on line 60: “veteran’s” should be ‘veterans’ Please add a comma between the ‘17’ and ‘18’ in line 166. Please remove the space before the period in line 236

Response: The manuscript was reviewed for copyediting/consistency issues and applicable revisions were incorporated, including the specific revisions noted in Reviewer #1’s recommendations. 

3. I was unclear why the paragraph beginning on line 98 was part of the introduction. It describes the particulars of what you did in this study and your statistical approach - shouldn’t it be in the method section? 

Response: Thank you for this comment, we changed this last introduction paragraph to focus on why we conducted this study and a big picture review of the work we present and our hypotheses. 

4. The sentence related to line 137 was unclear regarding if the ‘number of deployments’ was the only characteristic assessed “unique to military and veteran populations”. If so, please change “characteristics” “characteristic” and if not please add an e.g. before “number of deployments”.

Response: We collected various variables regarding military service and veteran status. We edited the sentence as follows:

“Participants were first asked to describe characteristics unique to military and veteran populations (e.g., number of deployments)…”

5. Was the survey anonymous? I assumed it did not collected identifying information as collecting names/email addresses was not stated anywhere (as is common with web based surveys). However I was unsure how individuals who reported “previous and current suicide ideation” (see line 171-172) were contacted. Please clarify this in the method section. Furthermore, I assume but am not sure if you meant “previous and/or current suicide ideation” based on your current language it appears that they would need both previous and current ideation to be contacted while those with just current would not be contacted. I am not sure this was your intent. Also, it may be useful to add the number of individuals contacted for this reason.

Response: The survey did collect e-mail addresses during the implied consent so we could implement the mitigation strategy for those who reported suicidal ideation. No other identifiers were collected. We clarified this in lines 174-75. “The only identifier collected was an email address to provide a mitigation strategy for those who reported prior suicidal ideation or attempt (see below).

Emotional/Behavioral health Domain section: “Participants who reported previous and/or current suicidal ideation, were contacted by the research team by e-mail…”

Original sentence: “Participants who reported previous and current suicidal ideation, were contacted by the research team “

6. Was the ‘loss of self instrument’ validated? Due to its brevity I would appreciate more information about this instrument including other usage and perhaps test-retest if available? 

Response: The loss of self is a validated measure first published by Skaff et. al (1992). We added additional information about the “loss of self” in the “Social Function/Well-being Domain” section.

Skaff, M. M., & Pearlin, L.I. (1992). Caregiving: role engulfment and the loss of self. The Gerontologist, 32(656-64). 

7. What does it mean when you “identified the extent to which caregivers had been diagnosed” on line 191. Does that mean you asked them if they had such a diagnosis? I”m not sure ‘extent’ is the best word in this context. 

Response: We clarify this sentence and edited the statement as follows:

“We identified previously diagnosed health conditions by asking caregivers to provide information about their health history and identify conditions for which they received a diagnosis by a healthcare provider during the period of time since beginning their role as a caregiver.”

8. On line 318-319 - “Suicidal ideation rates in this sample of military caregivers were higher than rates in other published studies measuring suicide-related behavior among family caregivers”

Please add a discussion about this - do you think this was pure chance or a real effect? Perhaps a change over time or due to different research methods? might this be due to your sampling method? What was stated in the advertisements that may affect this?

Response: The study recruited and enrolled military and Veteran caregivers from across the US using all available resources. Although a selection bias may have played a role, this may not have been associated to the advertisement since it was advertised as understanding the health and well-being of military and Veteran caregivers. We added the following sentence in the discussion section,

“There is limited existing literature on suicide in caregivers in the general population, and mostly when caring for a loved-one with mental health. Alzheimer and Dementia. When compared with studies addressing caregiving and suicide in the general population, this sample of military caregivers exhibited higher rates of suicidal ideation. This marked difference could be attributed to various factors, one could argue that military caregivers perform a higher complexity of care at a younger age, during the prime of their adulthood. (32-34) We recommend additional epidemiological studies to identify and understand caregiving and suicide in military and civilian caregivers.” 

9. Regarding the explanation beginning on line 347, there are many studies showing how identity disturbances can be related to developing suicidality. It might be worth using/citing some here. 

Response: Please see response to #8 above. We added a set of references to support this statement.

10. On line 364 you state that “the potential selection bias that resulted from the study eligibility requirement for participants to self-identify as a caregiver was mitigated by the development of an internal algorithm that identified the responses provided and compared them to unique characteristics meeting the definition of caregiving.” I did not understand what this meant. What is this internal algorithm? How was it used to help with potential selection bias? I don’t remember seeing this in your method. Please add a full explanation of this to your method section.

Response: We added the following sentence in the method section:

“The participants of this study self-identified as a military and Veteran caregiver. Each survey was evaluated based on the information provided regarding the caregiver role, performed tasks, period of time of caregiving as well as the care recipient military service and medical history. These variables served as a verification of caregiving status, specifically as a caregiver of a wounded, ill and injured military or Veteran.” 

11. I also was unsure your other reason to limit the issue of selection bias was presented in a satisfactory fashion. While your study sample characteristics may have been similar to the RAND study, they did not match other studies levels of suicide ideation (a central part of your results and discussion). You should note this in the context of how it may relate to selection bias as a limitation. Personally, I do not find this to be a major methodological issue or concern but this issue should be noted and left to the reader to determine if they consider this a critical limitation regarding the interpretation of the suicide findings. 

Response: Thank you! To date, the RAND study is the only military and Veteran caregiver population-based study. The comparison with the population in the RAND study was intended to depict the unique socio-demographic characteristics of military and Veteran caregivers, which has been established in the literature as being significantly different from civilian caregivers. Age is one of the most distinct differences, which translates to many other sociodemographic factors. We appreciate your constructive feedback and we revised the sentence referring to the comparison with the RAND study in the Discussion section.

“To date, the RAND study is the only population-based data in military and Veteran caregivers, which has been established in the literature as significantly different from caregivers in the general population. A comparison of the characteristics in this study’s sample to those reported by the RAND study (average age, education, and health-related outcomes such as mental health symptoms of depression, stress, and anxiety) suggest that they are remarkably similar. However, the RAND study didn’t study suicide ideation among caregivers, and with no other studies exploring the topic of suicidal ideation in military and Veteran caregivers, we won’t be able to compare with a similar sample of caregivers.”

AARP. (2015). Caregiving in the United States 2015. Retrieved from http://www.aarp.org/ppi/info-2015/caregiving-in-the-united-states-2015.html

12. Line 381 - I found the sentence beginning with ‘existing programs’ confusing. The two parts do not seem linked well. Also there was a typo - please remove the capital ‘I’ 

Response: Thank you for bringing this to our attention. We reviewed the sentence and it now reads as follows:

“To date few programs are addressing suicide in military and Veteran caregivers. This study shed light on some of the factors and caregiver characteristics associated to suicidal ideation, a contribution we hope can be incorporated in future prevention initiatives.”

Original sentence: “Existing programs and services for military caregivers may not be addressing suicide ideation among military caregivers, but understanding the factors that contribute to suicidality is important for prevention.”

13. Finally, when reading the results I was hoping for a discussion of some of the more unexpected findings such as why are shorter travel distances to a VA medical facility associated with high distress phenotype and why risk for alcohol usage was associated with the lower distress phenotype. Please add some explanation for these findings to the discussion. 

Response: Thank you for this suggestion. We believe that shorter travel distance may be related to confounding by indication in that people caring for Veterans with more severe mental health symptoms (e.g., depression/suicidality) may want to live closer to a VA to make it easier to obtain urgent care with clinicians who are familiar with their care rather than a community emergency room where continuity of care may not be available. The increased burden of this care was demonstrated in the distress phenotypes, which is one possible explanation for this finding. We added a statement regarding this in the methods.

Why caregivers in the low-distress group were more likely to report risky alcohol use is less clear. We hypothesize that alcohol use may be a primary stress management approach that leads to less perceived stress, less depressive symptoms, lower loss of self in addition to lower risk for suicide ideation or attempt. 

14. When reading table 1, I was unable to determine what the stars and p-values indicated. Ie what analyses were run on which variables that led to the p-values. Please clarify this

Response: We clarified the table legend. It now reads as follows,

 Note: differences between the phenotypes **p<=0.001, *p<.01

15. Figures need APA style labels - Figures are also unclear with text illegible. I was unable to read most of the text in the first figure and some of the text in the latter two figures.

Response: Thank you for this suggestion. We reviewed the figures to enhanced them. We are using the PLoS One “Vancouver” style. We reformatted the figures to follow the guidelines provided at https://journals.plos.org/plosone/s/figures. 

Reviewer #2

 This study is one of the first studies to examine suicide symptoms in military care giving population; highlighting a significant problem that has gained minimal attention.

The authors conducted an online survey targeted to military caregivers and reached approximately 400 individuals. Using GLVM, they determined 3 phenotypes of caregiver profile contributing to suicidal ideation.

The article is well written, and background section for the most part is synthetic and detailed.

The following items impacted my enthusiasm and if addressed will strengthen the manuscript.

1) Theoretical model: The model (figure 1) for caregivers experience map looks like a flyer and does not particularly correspond to the abbreviated description in the article. The figure is much more detailed with content that is not elaborated elsewhere and it is not clear how the choice of survey parameters is derived from this.

Response: The Military and Veteran Caregiver Experience Map was intended to show some of the factors that may serve as interference and pain points in the trajectory of caregiving in military and veteran caregivers, which is part of the premises of the fluid vulnerability theory. We recognize that the figure is not clear and even after trying to work on a product that could be in print, it was challenging given the time we have to submit these revisions. We added the following in the introduction section:

“The Military and Veteran Caregiver Experience Map depicts the factors that contribute to caregiver trajectory and the dynamics across a span of time and events in their caregiving journey”.

The editorial committee also had some constructive comments regarding this figure and despite having the authorization to use it and working with the copyright holders to enhance it, we decided to delete the figure and leave the citation for the readers to access electronically. 

2) Survey instrument: The survey is anonymous and not particularly well described. How was it developed? Was it beta tested? An internal algorithm is mentioned that “identified the responses provided and compared them to unique characteristics meeting the definition of caregiving” Please elaborate what this is and how it works. How does the survey know if the caretaking is of a Veteran or military personnel? While the survey is anonymous, there is outreach if the participant notes suicidal ideation, past or present. How is this accomplished if survey is anonymous?

Response: Thank you for these questions. We added the following information in the methods section: 

“Upon approval by the Institutional review board (IRB) at The University of Texas Health Science Center at San Antonio (UT Health San Antonio), we convened a “Military and Veteran Caregiver Advisory Group” and engaged the support of national organizations serving military caregivers for distribution of our web-based survey.

The Military and Veteran Caregiver Advisory Group consisted of ten caregivers, caring for a wounded, ill, and injured veteran: spouses (4) between the ages of 30 and 60 years old, and parents (3) older than 55 years old, caring for a veteran of the wars in Iraq and Afghanistan (post 9/11) and spouses of veterans (3) of the wars before Iraq and Afghanistan (pre 9/11). Figure 2 provides a snapshot of this advisory group. Members represented the various characteristics of military caregivers in the community (i.e., pre-and post-9/11, length of caregiving time, myriad of injuries) across the United States[15]. The advisory group provided recommendations for the survey that allowed us to address the diversity of caregivers in the community and assisted with the testing of the web-based survey instrument. Goals of the instrument testing were to measure the average time of completion, content, language, and order of the questions/instruments. 

The advisory group also served as a facilitator in recruiting participants from interest groups related to their caregiving focus. We provided a link to the survey through social media, newsletters, flyers, and e-mail from national nonprofit organizations serving this population (e.g., Hearts of Valor, The Elizabeth Dole Foundation, Military Veteran Caregiver Network and others). Caregivers responded to web-based links in social media or e-mail information from caregiver organizations. Caregivers were not compensated for their time. To increase the response rate, survey length was limited to allow for completion within 30 minutes.”

3) Participant concerns:

Given that the recruitment includes VA Caregivers Program participants (48%), these individuals are often compensated for their involvement with the Veteran. This seems fundamentally different than caregivers who do not receive financial support. How is this captured in your design and findings?

Response: Thank you for this feedback. We conducted an analysis to explore any differences between the two groups of caregivers based on VA compensation. The results are below. We added a sentence in the results section.

VA Compensation High Low Medium

No 57 57 118

Yes 35 52 132

NA 1 4 0

 Pearson's Chi-squared test

X-squared = 5.9044, df = 2, p-value > 0.05

4) The participants are 85% Caucasian and <2% African American. This does not represent Veteran population nor I suspect Veteran caregivers. This is a limitation and should be mentioned. Is this due to recruitment outreach methods? What about bias of who is willing to complete survey with no compensation?

Response: Correct. The population data from the RAND Report and AARP studies reported the population as follows,

Caucasian (white) approximately 75% and Black (non-Hispanic) approximately 10%. We added the following in the discussion section: 

“Minority groups in the sample of this study were underrepresented. This is a limitation that could have resulted from various factors to include, recruitment strategies. This limitation may play a role in the rate of behavioral health conditions reported in the sample, typically being higher in minority groups”.

5) Statistics figures are hard to read and poor quality. 

Response: We apologize for this. We formatted the figures and created new ones to enhance the quality and clarity.

6) Limitations: The study focuses solely on symptoms coincident with caretaking responsibilities and therefore there is no data on pre-existing problems, or even prior suicidal symptoms. The introduction refers to the fluid vulnerability theory making the argument for static (e.g. biological) and dynamic processes, but this paper ignores pre-existing conditions. This needs to be acknowledged in the limitations section. What was the rationale for not asking about previous suicide symptoms/history, given the strong connection about past suicidal symptoms being significant risk factor for repeated occurrence?

Response: We agree with you and we should have noted this, so we have addressed this in the (revised) discussion section. The study sought to identify suicidal ideation since the time the participant became a caregiver, but we did not ask the caregivers for pre-existing conditions. We added the following in the discussion section. 

“The limitations of this study…. The study did collect information of the caregiver pre-existing conditions, especially those associated with behavioral health that could have an association with suicidal ideation”

7) Typo – line 382

Response: Thank you for noticing. We corrected the typo.

---

## [Decision Letter · Decision Letter 1]

31 May 2021

Phenotypes of caregiver distress in military and veteran caregivers: suicidal ideation associations

PONE-D-21-01599R1

Dear Dr. Delgado,

We’re pleased to inform you that your manuscript has been judged scientifically suitable for publication and will be formally accepted for publication once it meets all outstanding technical requirements.

Kind regards,

Gianluigi Forloni

Academic Editor

PLOS ONE

Additional Editor Comments (optional):

Reviewers' comments:

Reviewer's Responses to Questions

**Comments to the Author**

1. If the authors have adequately addressed your comments raised in a previous round of review and you feel that this manuscript is now acceptable for publication, you may indicate that here to bypass the “Comments to the Author” section, enter your conflict of interest statement in the “Confidential to Editor” section, and submit your "Accept" recommendation.

Reviewer #1: All comments have been addressed

2. Is the manuscript technically sound, and do the data support the conclusions?

Reviewer #1: Yes

3. Has the statistical analysis been performed appropriately and rigorously? 

Reviewer #1: Yes

4. Have the authors made all data underlying the findings in their manuscript fully available?

Reviewer #1: (No Response)

5. Is the manuscript presented in an intelligible fashion and written in standard English?

Reviewer #1: (No Response)

6. Review Comments to the Author

Reviewer #1: Thank you for revising the manuscript in line with the suggestions made by myself and the other reviewer.

7. PLOS authors have the option to publish the peer review history of their article (what does this mean?). If published, this will include your full peer review and any attached files.

Reviewer #1: No

---

## [Editor Report · Acceptance letter]

4 Jun 2021

PONE-D-21-01599R1 

Phenotypes of caregiver distress in military and veteran caregivers: suicidal ideation associations 

Dear Dr. Delgado:

I'm pleased to inform you that your manuscript has been deemed suitable for publication in PLOS ONE. Congratulations! Your manuscript is now with our production department. 

Kind regards, 

on behalf of

Dr. Gianluigi Forloni 

Academic Editor

PLOS ONE